# Real Life Prospective Evaluation of New Drug-Eluting Platform for Chemoembolization of Patients with Hepatocellular Carcinoma: PARIS Registry

**DOI:** 10.3390/cancers12113405

**Published:** 2020-11-17

**Authors:** Thierry de Baere, Boris Guiu, Maxime Ronot, Patrick Chevallier, Géraldine Sergent, Illario Tancredi, Lambros Tselikas, Marco Dioguardi Burgio, Lucas Raynaud, Frederic Deschamps, Gontran Verset

**Affiliations:** 1Department of Interventional Radiology, Gustave Roussy Cancer Center, 114 rue Edouard Vaillant, 94805 Villejuif, France; lambros.tselikas@gustaveroussy.fr (L.T.); frederic.deschamps@gustaveroussy.fr (F.D.); 2UFR Médecine Le Kremlin-Bicêtre, Université Paris-Saclay, 94250 Le Kremlin-Bicêtre, France; 3Imagerie Médicale St Eloi, 34000 Montpellier, France; b-guiu@chu-montpellier.fr; 4Service de Radiologie, Hôpital Beaujon, APHP Nord, Clichy & Université de Paris, 92210 Clichy, France; maxime.ronot@aphp.fr (M.R.); marco.dioguardiburgio@aphp.fr (M.D.B.); lucas.raynaud@aphp.fr (L.R.); 5Service d’Imagerie Médicale Diagnostique et Interventionnelle, Hôpital L’Archet II, 06200 Nice, France; chevallier.p@chu-nice.fr; 6Radiologie & Imagerie Digestive, CHRU Lille, 59037 Lille, France; geraldine.sergent@chru-lille.fr; 7Gastroentérologie Médicale, Hôpital Erasme, Université Libre de Bruxelles, 1070 Brussels, Belgium; illario.tancredi@chu-charleroi.be (I.T.); Gontran.Verset@erasme.ulb.ac.be (G.V.)

**Keywords:** hepatocellular carcinoma, DEM-TACE, doxorubicin, idarubicin, hepatobiliary toxicities

## Abstract

**Simple Summary:**

Hepatocellular carcinoma treatment options depend on stage of disease. In intermediate stage transarterial chemoembolization with drug-eluting microspheres (DEM-TACE) is recommended. DEM-TACE is simultaneous embolization of tumour feeding arteries and local delivery of anticancer drugs. We assessed real-life practice, safety, toxicity and efficacy of this therapy using new embolization microspheres in 97 patients. Toxicity of the treatment in our study was within or below rates reported so far, and the healthy liver parenchyma, the bile ducts and the portal vein were well preserved when compared with previous study using other type of DEM. Tumour response rate was high, achieving disease control in almost all patients. Hepatocellular carcinoma was controlled during 16.7 months with DEM-TCE as the only treatment. At one year 81% and at two years 66% of patients were alive. Our study showed that DEM-TACE in patients from every-day clinical practice is safe and efficient treatment modality.

**Abstract:**

Background and aim: Transarterial chemoembolization with drug-eluting microspheres (DEM-TACE) is recommended for patients with BCLC stage B hepatocellular carcinoma (HCC) and stage 0-A unsuitable for curative treatments. We assessed efficacy and safety along with hepatobiliary toxicities (HBT) of DEM-TACE using a novel microsphere, LifePearl^TM^, loaded with anthracyclines. Materials and methods: 97 patients diagnosed with HCC were prospectively enrolled and treated using LifePearl^TM^ loaded with doxorubicin (77%) or idarubicin (23%). Safety and tolerability were assessed using CTCAE, HBT by CT/MRI scans, and tumor response by applying modified Response Evaluation Criteria in Solid Tumors (mRECIST). Follow-up was after 2 years. Results: Adverse events (AE) were reported in 73.2% of patients, majority being Grade 1–2. Grade ≥ 3 AE reported in 13.4% of patients were mainly related to postembolization syndrome. HBT were observed after 15.5% (29/187) of the DEM-TACEs. Objective response and disease control rates were 81% and 99%, respectively, as the best responses. Survival rates at one and two years were 81% and 66%, respectively, while the median overall survival (OS) was not reached. Median progression free survival was 13.7 months (95% CI: 11.3; 15.6) and median time to TACE untreatable progression was 16.7 months (95% CI: 12.7; not estimable (n.e.)). Conclusions: DEM-TACE using LifePearl^TM^ provides a high tumor response rate in HCC patients. HBT rates within or below previously reported results for cTACE and DEM-TACE indicate a good safety profile for LifePearl^TM^. The trial was registered in National Library of Medicine (ID: NCT03053596).

## 1. Introduction

Hepatocellular carcinoma (HCC) is among the six most common types of cancer in the world and the third leading cause of cancer death [1], often associated with concomitant liver impairment [2,3]. Patient stratification and treatment allocation are based on factors related to tumor burden, liver function and performance status, according to the Barcelona Clinic Liver Cancer (BCLC) staging system which is commonly used for HCC management in Europe and the United States [2,3,4,5]. Resection, liver transplantation, and local ablation are considered potentially curative in carefully selected patients, with 5-year survival rates of 40–70%, compared with 20% in untreated patients [3,4,5]. Following the BCLC staging system and treatment strategy, transarterial chemoembolization (TACE) is the first-line treatment for intermediate-stage disease, a possible option to maintain a patient on the waiting list or to downstage a patient for liver transplantation and can also be used according to treatment stage migration for early-stage disease when curative treatments are not feasible or failed [2,5]. There are two TACE techniques: conventional TACE (cTACE) and TACE with drug-eluting microspheres (DEM-TACE). DEM-TACE allows the delivery of high concentrations of chemotherapeutic drugs inside the tumor, with low systemic exposure and high tissue necrosis due to durable ischemic effect of the embolic material inside the tumor-feeding vessels [2,4]. TACE is generally recognized as a relatively safe therapy with a low rate of severe adverse events. Recently, few retrospective studies assessed liver and biliary injuries on TACE imaging follow-up and concluded that bile duct, portal vein or liver parenchyma damages are not rare and are more frequent with DEM-TACE than with cTACE [6,7].

Several types of drug-eluting microspheres are currently available with different characteristics such as drug elution profile, size distribution, stability in suspension, chemical and physical properties [8]. LifePearl™ (Terumo Europe, Leuven, Belgium) is made of polyethylene-glycol, a hydrophilic material that allows good compressibility, elasticity, and maximizes the time in suspension [8]. Several studies report good safety and efficacy of LifePearl™ when used for DEM-TACE to treat both primary and secondary liver tumors [9,10,11,12,13,14]. LifePearl™ can be loaded with various anthracyclines, such as doxorubicin, epirubicin and idarubicin, for DEM-TACE in the treatment of HCC. The data on safety and efficacy of idarubicin started to emerge only in recent years [15,16]; epirubicin is the preferential drug in some countries, while doxorubicin remains the most used drug for HCC treatment. The main purpose of this study was to assess tolerance and toxicity (hepatobiliary and systemic), safety and tumor response after DEM-TACE with LifePearl™ microspheres loaded with anthracyclines in a real-life setting.

## 2. Results

### 2.1. Patient Population

The PARIS registry enrolled 97 patients that were affected by unresectable HCC and treated with LifePearl™ loaded with doxorubicin or idarubicin. The baseline patient’s characteristics are presented in Table 1. BCLC A (early) and B (intermediate) stage were the most represented characteristics of HCC. Mean number of tumors per patient was 2.3 ± 1.6. Most of the baseline characteristics were similar in patients treated by doxorubicin and idarubicin except for cirrhosis that was higher in doxorubicin group (91.7% versus 59.1%; *p* = 0.0002) and for the sum of tumor diameters that were higher in idarubicin group (57.2 ± 35.7 mm vs. 84.4 ± 53.9 mm; *p* = 0.023), respectively. As expected, there was a significantly lower number of tumors and sum of diameters in BCLC 0&A, but also more female patients were present in this subgroup (13% vs. 2%; *p* = 0.04).

Previous treatments, before entering PARIS study, included 17 (18%) ablation, 9 (9%) surgery, 6 (6%) c-TACE, 3 (3%) DEM-TACE, 1 (1%) liver transplantation, and 2 (2%) selective internal radiation therapy.

Among the 97 enrolled patients, hepatobiliary damages were detected on baseline CT/MRI imaging in 10 patients (two bilomas, five portal vein thromboses, two portal vein branch narrowing, and one bile duct dilatation). Those 10 patients presented with larger tumors than the rest of the population with the median sum of tumor diameters of 100.5 mm (range 23.0–237.0) versus 51.5 mm (range 13.0–180.0) and 30% (3/10) were BCLC C. Three out of these 10 patients with hepatobiliary toxicities (HBT) at baseline had one or more prior locoregional treatments, while the remaining seven did not report any treatment before inclusion in the PARIS study.

### 2.2. Treatments

A total of 187 DEM-TACE were performed in 97 patients (25.7% with 100 μm and 74.3% with 200 μm LifePearl™); 75 patients (77%) were treated with doxorubicin and 22 (23%) with idarubicin. The mean administered dose per DEM-TACE was 74 ± 17 mg (range 25–150 mg) for doxorubicin and 12 ± 4 mg (range 5–20 mg) for idarubicin. A single treatment was performed in 45.4% (44/97) of patients, while 54.6% (53/97) of the patients received at least two DEM-TACE treatments (Table 2).

Mean number of treatments per patients was 1.93 ± 1.14 (range 1–7), significantly lower in patients treated with idarubicin as compared to doxorubicin (1.4 vs. 2.1; *p* = 0.001). Not surprisingly, significantly more patients in BCLC B&C stage had bilobar treatment as compared to BCLC 0/A (49.0% vs. 17.4% *p* = 0.001) (Table 2). Complete embolization of the tumor-feeding artery of target lesions was achieved in 95% of the DEM-TACEs. Spasm in the tumor-feeding arteries either from the liver or collaterals (i.e., phrenic artery) were the main reason for incomplete embolization of the target tumors.

### 2.3. Safety and Toxicity

Adverse events (AE) were reported in 71/97 (73.2%) patients. Most frequent AE were post embolization syndrome, with AE intensity grade (G) 1–2 in 71.0% of cases. G ≥ 3 adverse events were reported by 13/97 (13.4%) patients and were mainly related to postembolization syndrome (Table 3).

One patient with a history of heart insufficiency, with 65% liver replacement and the largest tumor measuring 20 cm, died from heart failure two days after DEM-TACE performed with 4 mL of LifePearl™ loaded with 5 mg/mL of idarubicin, despite a successful procedure, without postembolization syndrome.

There was no report of any adverse event related to changes in laboratory parameters. A mild, transient increase in ALT, AST and bilirubin was observed in 10.3%, 7.2% and 6.2% of patients, respectively.

### 2.4. Hepatobiliary Toxicities

The analysis of 366 follow-up imaging exams (CT or MRI) revealed development of nine bilomas, nine portal vein thromboses (PVT), two portal vein branch narrowing (PVBN) and nine bile duct dilations (BDD) resulting in hepatobiliary toxicities (HBT) after 15.5% (29/187) DEM-TACE. The rate of hepatobiliary toxicities (HBT) was not significantly different when using LifePearl™ loaded with idarubicin (10/31, 9.7%) or with doxorubicin (26/156, 16.7%; *p* = 0.58).

In three patients, one with PVT, one with PVBN and one with biloma, damages were seen at first follow-up imaging and then resolved.

Ten of the 16 patients (62.5%) with one or several prior liver-directed therapies (LDT) (four surgical resections, six thermal ablations, five cTACE, two each DEM-TACE and radioembolizations) developed HBT. They account for 33% of the 29 patients that had HBT. Among the 68 patients that did not develop HBT, only 6 (8.8%) had prior LDT (*p* < 0.01).

The patients with HBT had a mean of 2.4 ± 1.4 (median 2; range 1–7) DEM-TACE, and HBT occurred after more than one course in 66% of the patients. Three patients developed biloma after the first DEM-TACE, three after the second and three after 3 to 7. In three patients, all with multiple TACE treatments, 2 or 3 concomitant damages were observed on the same follow-up imaging. After development of HBT, 62% of patients did not receive any additional TACE while those remaining were subsequently treated by one to three DEM-TACE treatments. Patients who had hepatobiliary damages at enrollment in the study had a median of one treatment (range 1–5). The strongest predictor for HBT was previous locoregional treatment with the OR 2.65 (95% CI: 1.07–6.61) (Figure 1).

There was no significant difference in progression-free survival (PFS) (median; 95% CI): 15.2 (10.3; 17.0) vs. 13.4 (9.9; 15.6) *p* = 0.44), time to TACE untreatable progression (TTUP) (median: 16.5 (11.4; not estimable (n.e.)) vs. not estimable (11.7; n.e.) *p* = 0.98), or overall survival (OS) (median: 22.5 (95%CI: 16.2; n.e.) months vs. not-estimable; *p* = 0.30) between patients with or without HBT, respectively (Figure 2).

### 2.5. Tumour Response and Progression-Free Survival

Modified Response Evaluation Criteria in Solid Tumors (mRECIST) assessment after DEM-TACE in target lesions showed an objective response rate (ORR) of 80.8% and disease control rate (DCR) of 98.9% as the best response. ORR and DCR were 76.4% vs. 95.2% and 89.7% vs. 100% in doxorubicin and idarubicin, respectively; they were 84.4% vs. 77.5% and 97.7% vs. 100% in BCLC 0&A and BCLC B&C, respectively (all differences not significant). When considering both target and non-target lesions, the ORR was 72.9%, and the DCR was 91.4%.

The median overall PFS was 10.6 (95%CI: 7.9; 12.4), target lesion PFS was 13.7 months (95%CI: 11.3; 15.6), while the median TTUP was 16.7 (95%CI: 12.7; n.e.) months (Figure 2). The median PFS in doxorubicin and idarubicin groups was 15.7 (95%CI: 11.4; 20.4) and 10.4 (95%CI: 6.3; 14.3) months, respectively (*p* = 0.015), while it was 15.2 (95% CI: 11.3; 22.5) and 12.7 (95% CI: 8.2; 17.0) months (*p* = 0.20) for BCLC 0&A and BCLC B&C, respectively.

Patients with a complete response (CR) had significantly longer median TTUP than patients who did not reach CR (n.e. (95%CI: 13.6; n.e.) vs. 15.1 (95%CI: 12.0; n.e.); *p* = 0.015).

Biologic tumor response evaluated with alpha-fetoprotein (AFP) serum level was available before index treatment and at follow-up in 43/97 (44%) patients with a mean baseline value of 330.6 ± 1026.6 (range 2–5700) ng/mL and matched follow-up values of 92.0 ± 241.8 (range 2–1200). The ratio of changes (x-times increase or decrease) between baseline and first follow-up is shown in Figure 3.

### 2.6. Overall Survival

Survival rates at one and two years were 81% and 66%, respectively, while median overall survival was not reached (Figure 4). Median OS was not reached in BCLC 0&A, while it was 22.2 months (95%CI: 16.2; n.e.) in BCLC B&C patients (*p* = 0.007).

Patients with target lesion complete response (CR) as per mRECIST had significantly longer median overall survival compared to patients who did not reach CR (not reached vs. 22.4 (95%CI: 15.8; n.e.)), respectively (*p* < 0.001).

Cox regression model identified several predictors for OS, such as, three or more comorbidities with hazard ratio (HR) and 95%CI of 3.49 (1.72, 7.08), highest dose of anthracyclines (≥ 150 mg doxorubicin or ≥20 mg of idarubicin) HR 3.58 (1.46, 8.72), CR of target lesions HR 0.337 (0.155, 0.734), as indicated in Figure 5.

Fourteen patients (14.4%) underwent successful liver transplantation, two (2.1%) were successfully down-staged by DEM-TACE and underwent hepatectomy, and two (2.1%) patients were lost to follow-up.

## 3. Discussion

DEM-TACE was introduced to optimize and standardize long-used cTACE, by ensuring sustained and prolonged release of therapeutic agents supplemented by embolization that are expected to act synergistically. The superiority of one TACE technique versus another has not been established [17,18], and both techniques hold the highest level of recommendation for intermediate stage HCC in various guidelines [2,3] based on overall survival benefit of cTACE versus best supportive care [19,20]. In the last decade, several microspheres have been developed with the intent to further improve DEM-TACE. LifePearl™ microsphere has shown some favorable mechanical and pharmacological characteristics, including the release of a higher percentage of the loaded anthracyclines, sphericity, uniform size distribution and extended time in suspension [8].

The main findings of our study are that DEM-TACE using anthracycline-loaded LifePearl™ provides a high level of disease control for HCC, within or above the rate previously reported in the literature. This high response rate has been obtained irrespective of the type of anthracycline or stage of disease, with acceptable safety. Global HBT rate was within or below the previously reported range [6,7]. Another interesting finding of this real-life prospective study is the high proportion of patients in the early stage of disease (BCLC 0 and A) receiving TACE, thus confirming the trend of using DEM-TACE as a bridge to transplant or as downstaging to resection [21]. Understanding and minimizing the toxicity of TACE is nowadays of utmost importance in order not to obviate possibilities of further treatment including curative, surgery or transplantation, as well as highly promising, first line systemic combination immunotherapies [22].

In the present study, HBT were encountered after 29 of 187 (15.5%) procedures, which is a lower rate than previously reported for DEM-TACE of HCC, and within the range reported for cTACE [6,7]. Monier et al. reported HBT after 14.4% of procedures with cTACE and 36.8% with DEM-TACE [7]. Guiu et al. reported HBT after 4.2% for cTACE and 30.4% for DEM-TACE in a population of cirrhotic patients, while reporting that hepatobiliary injuries were significantly more frequent in the non-cirrhotic liver [6]. It is noticeable that in our study, 16% of the patients did not have cirrhosis (thus maximizing the probability of HBT), and that imaging follow-up was up to 20 months (thus maximizing the likelihood to detect HBT). Lower incidence of HBT in our study than in studies with first generations of DEM [6,7] allows us to hypothesize that DEM-TACE has been optimized through time, by the refinement of the techniques and devices such as 3D vascular navigation, innovative catheters and improved mechanical properties of microspheres that synergistically reduce non-target embolization and drug toxicity. Those improvements are particularly relevant for patients with multiple treatments, like in our study. The long-lasting embolic effect of microspheres could prevent re-access to tumor-feeding arteries during subsequent treatments and increase the chance for non-target embolization and consequent injury of peribiliary plexus when DEM reach the arterial supply of healthy liver parenchyma during repeated TACE. Such an increase in non-target embolization towards healthy liver might be supported by the mean of 2.4 ± 1.4 DEM-TACE in patients suffering from HBT, and by the fact that 66.0% of HBT occurred after more than one course of DEM-TACE. The development of drug loadable bioresorbable microspheres might overcome this limitation of DEM-TACE in the future. Hepatic changes induced by previous therapies, particularly recent ones, were reported to have a significant impact on the occurrence of HBT [23,24]. Such a finding was confirmed in our study by logistic regression analysis, but also by the development of HBT in 10 of the 16 (62.5%) patients that previously received 17 LDTs for HCC.

An interesting observation in our study is the trend towards lower HBT rate in patients treated with idarubicin versus doxorubicin, despite significantly less patients with cirrhosis. Both drugs are known for their strong vesicant effect, but whether it is the magnitude of the vesicant effect, the difference in the dose administered, the lower number of DEM-TACE or the effect of chance remains to be confirmed in larger-scale trials.

Clinical relevance of HBT is still under discussion. A logistic regression analysis did not identify HBT as a predictive factor for OS, and OS, PFS and TTUP did not differ between patients with or without HBT in this study. Furthermore, HBT were reported as adverse events in only two patients, the remaining having been identified by imaging alone. Different pathophysiological mechanisms are proposed as triggers for the development of HBT but, to date, predictive factors (either patient or DEM-TACE related) have not been clearly identified. Consequently, there is no clear guidance to prevent them. In the present study, seven patients with hepatobiliary damage at baseline did not have any history of prior LDT, indicating that other factors also play a role in their development.

Median OS was not reached in the current study. This finding is consistent with previous studies reporting some of the longest survival times of patients in this stage of the disease. Burrel and colleagues [25] reported 89.9% and 66.3% survival rates at 1 and 3 years, respectively, whereas Ikeda and colleagues [26] reported a median OS of 3.1 years and 75% survival rate at 2 years. However, studies with a less selected, real-world population, reported two years survival of 40.0% to 55.9% [18,27,28]. Cox regression analysis identified pre-existing multiple comorbidities as a negative predictor for OS with HR 3.48, reflecting unselected patient enrollment. Potassium and alkaline phosphatase (ALP) were also significant negative predictors of survival. Potassium abnormalities are associated with poorer outcomes, particularly in patients with cirrhosis ascites and renal impairment. ALP, although present in many tissues, also indicates proliferation of the tumors and has been described as a negative predictor for survival of patients with HCC after hepatic resection [29]. This finding agrees with higher dose of anthracyclines as another negative predictor because a higher dose is needed in case of larger or multiple tumors. Target lesion complete response was a favorable predictor with HR 0.34. Noticeably, complete response after third or fourth TACE was achieved in 20% of the patients. This finding indicates that, in patients with preserved liver function, TACE is still valuable beyond two treatments. In this context, TTUP with TACE appears to be an interesting endpoint as it highlights the duration of disease control with DEM-TACE. TTUP in the current study had a median of 16.7 months and many patients were switched to other therapies without reaching unTACEable condition.

The response rate observed in our study is among the highest reported ones, with an objective response rate of 81% and a disease control rate of 99%. High OR and DC rates were observed with both idarubicin and doxorubicin. Our findings are similar to previously reported results for LifePearl™ with best OR rate ranging between 76% and 100% [9,10,11,12,13,14], and even superior to other studies that reported an OR rate from 52% to 74.8% [17,18,27,28,30].

The results of our study confirm the importance of obtaining a high response rate and of aiming for CR, even if several DEM-TACEs are necessary. We demonstrate that it offers longer TTUP and longer OS. Targeted research addressing subsequent TACE until CR is obtained, in a well-defined population, should bring more clarity on the risks–benefits of that approach.

The 13.7 months median PFS in the current study was relatively long, with no significant differences according to BCLC stages. PFS in the doxorubicin group (15.7 months) was significantly longer than in the idarubicin one (10.4 months). The difference in PFS could be explained by significantly larger treated tumors in the latter group. However, it remains longer than the median 6.6 months PFS reported by Guiu and colleagues with idarubicin, and longer than the 7.7-month PFS reported by Roth and colleagues when comparing idarubicin and doxorubicin [31,32]. TACE with idarubicin has gained a lot of interest in recent years and the results obtained in our study are among the best reported so far [15,16,31,32].

The main limitations of this study are the absence of a control arm, unselected patient population with multiple comorbidities and treatments following routine clinical practice, resulting in variability of the patient management, timing of imaging follow-up and response evaluation. However, inclusion of all patients eligible for TACE in this study and liberal assessment protocols give a realistic picture of contemporary TACE practice and patient outcomes.

## 4. Materials and Methods

### 4.1. Study Design and Outcome Measures

The PARIS registry was an international, multicenter, single-arm, prospective study conducted in six European centers (one in Belgium, five in France). Written informed consent was obtained from each patient included in the study and the study protocol conforms to the ethical guidelines of the Declaration of Helsinki as reflected in a priori approval by the institution’s human research committee. Approval from comité consultatif sur le traitement de l’information en matière de recherche and was obtained. The trial was registered in ClinicalTrials.gov National Library of Medicine (ID: NCT03053596).

Patients were included in the study if they fulfilled the following inclusion criteria: ≥18 years old, diagnosis of unresectable HCC, suitable for, and assigned to TACE treatment by a multidisciplinary tumor board, treated with LifePearl™ loaded with anthracycline and signed informed consent for participation in the study.

Primary outcome measures were safety, hepatobiliary toxicities (imaging finding of biloma, portal vein thrombosis, portal vein branch narrowing and bile duct dilatation), tumor response (assessed by mRECIST criteria); overall survival (OS); progression-free survival (PFS) defined as first observed disease progression or death; and time to TACE untreatable progression (TTUP) defined as time to last observed progression after prior, at least, disease control achieved by DEM-TACEs.

### 4.2. Patient Treatment

DEM-TACE was performed according to the clinical practice of each participating hospital concerning choice of microspheres size (100 mm or 200 mm), loaded drug (doxorubicin or idarubicin) or selectivity of embolization. Delivery of the microsphere followed LifePearl™ instruction for use and published recommendations [33]. The number of treatment cycles and re-treatments were on demand according to hospital practice and at the discretion of treating physicians. Further treatment options of patients that reached the TACE untreatable condition were at the discretion of the multidisciplinary tumor board.

### 4.3. Follow-Up and Assessments

Follow-up was according to routine hospital practice and scheduled for two years, TACE untreatable progression, or death. Patients who were resected, transplanted or reached TTUP were followed only for survival. Liver toxicity assessment was based on routine blood tests of liver enzymes (ALT—alanine aminotransferase; AST—aspartate aminotransferase), alkaline phosphatase (ALP), GGT (gamma-glutamyl transferase), complete blood count (CBC), coagulation tests, bilirubin and albumin and on their changes related to DEM-TACE treatment. Type and intensity of adverse events (AE) were monitored after each DEM-TACE treatment and throughout the follow-up of patients and were classified according to Common Terminology Criteria of Adverse Events (CTCAE) v4.03. All baseline and follow-up images were examined in the treating hospital for hepatobiliary toxicities as described elsewhere [6], including biloma, portal vein thrombosis, portal vein branch narrowing and bile duct dilatation. Tumour response was assessed according to modified Response Evaluation Criteria in Solid Tumors (mRECIST) [34].

Some subgroups of patients (doxorubicin vs. idarubicin; BCLC 0/A vs. BCLC B/C) were separately analyzed to assess treatment efficacy and hepatobiliary toxicities.

### 4.4. Statistical Methods

Patient demographics, medical history, disease characteristics and procedure parameters are presented with the mean ± standard deviation (SD), median and range, and 95%CI based on the t-distribution for continuous variables, frequencies and percentages with exact Clopper–Pearson 95% CI for discrete variables. Chi-square test was used for testing categorical variables, and the Wilcoxon rank-sum to compare continuous variables between subgroups.

Results of time-to-event endpoints are estimated using the Kaplan–Meier method and presented graphically as Kaplan–Meier curves along with a 95% two-sided confidence interval based on Greenwood formula, with Log Rank *p*-values calculated to test for differences between subgroups. Time to OS was calculated by subtracting the enrolment date from the death date or censor date (the date the subject was last observed and was alive). For PFS, the date of progression or the death date was used as event date. For TTUP, time to event was calculated by subtracting the index date from the date of last progression after the last disease control. Similar calculations were used to calculate time to best response, using the first date the overall best response was reached as the event date. For TTUP, PFS, and best response, times are censored at the last date an imaging was performed. A sensitivity analysis was performed, censoring the times at the date the subject was last observed or the date of resection/transplantation. Cox regression (CoxR) was performed to predict OS using the following predictor variables: age, gender, Child Pugh, ECOG, liver enzymes, blood parameters, alpha-fetoprotein (AFP), comorbidities (respiratory, cardiovascular, gastrointestinal, metabolic or endocrine, hematological, immunological and others) tumor burden, liver segment(s) involved, BCLC stage, hepatobiliary toxicities, number of procedures, dose of anthracyclines and complete response. To predict hepatobiliary toxicities, logistic regression (LR) modeling was used and the following variables were included: age, gender, comorbidities, cirrhosis, Child Pugh, ECOG, all blood parameters (including coagulation parameters), prior locoregional treatments, tumor burden, stage of disease, number of TACE, dose and type of anthracyclines. For both CoxR and LR models, a similar procedure was followed to create a multivariable model: in a first step, univariate CoxR and LR were performed on each predictor variable separately to assess the unadjusted hazard ratios or odds ratios with 95%CI. In a second step, all variables with a univariate *p*-value ≤ 0.10 were added into a multivariable model. In a final step, a stepwise Cox or logistic regression was performed on this selection, in which variables with an adjusted *p*-value ≤ 0.25 were sequentially entered into the model, whilst being retained if they demonstrated an adjusted *p*-value ≤ 0.10 in the subsequent steps of the stepwise selection. All analyses were carried out using SAS software, version 9.4 (The SAS Institute, Cary, NC, USA). All statistical tests were 2-tailed. The analysis between subgroups was exploratory (non-randomized comparisons) and therefore, *p*-values need to be interpreted accordingly.

## 5. Conclusions

DEM-TACE using LifePearl™ loaded with doxorubicin or idarubicin in a real-life setting proved to be a valuable treatment option for HCC patients with good tolerance, acceptable toxicity and a high and lasting tumor response that translated into satisfactory disease control and overall survival. CR was predictive of OS. Hepatobiliary toxicities were within or below the reported range for TACE, and local therapy of HCC before DEM-TACE seems to amplify HBT.

## Figures and Tables

**Figure 1 cancers-12-03405-f001:**
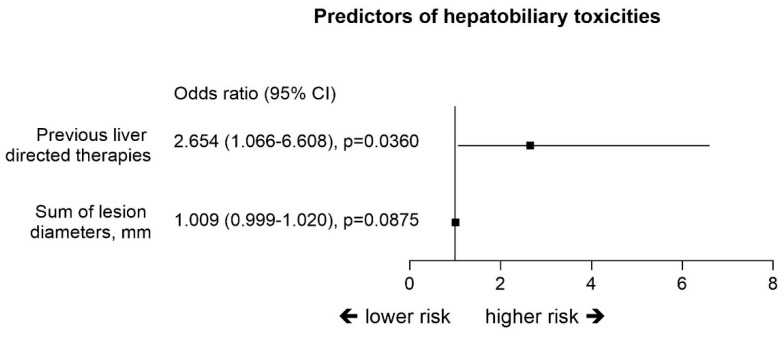
Multivariate analysis for the prediction of hepatobiliary toxicities (Logistic regression models).

**Figure 2 cancers-12-03405-f002:**
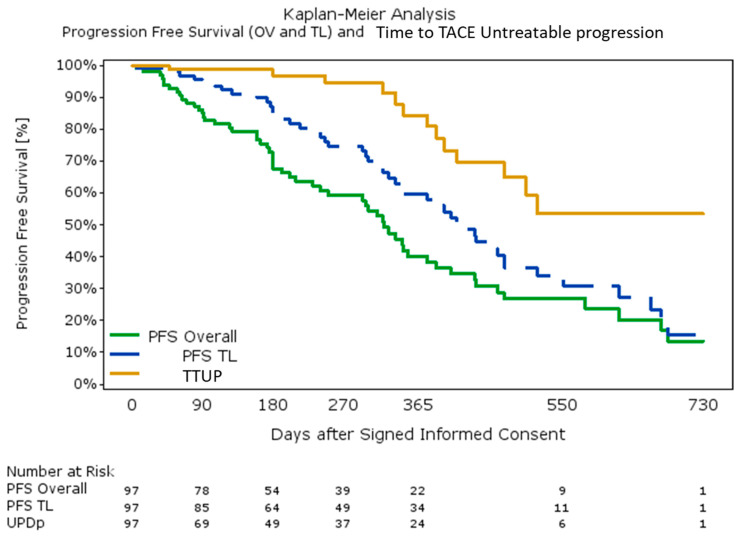
Kaplan–Meier Analysis: progression free survival (overall and target lesion) and time to transarterial chemoembolization (TACE) untreatable progression.

**Figure 3 cancers-12-03405-f003:**
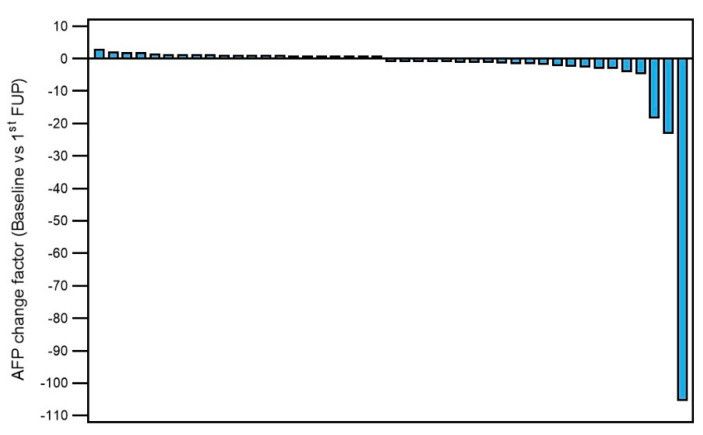
Alpha-fetoprotein (AFP) change between baseline and first follow-up.

**Figure 4 cancers-12-03405-f004:**
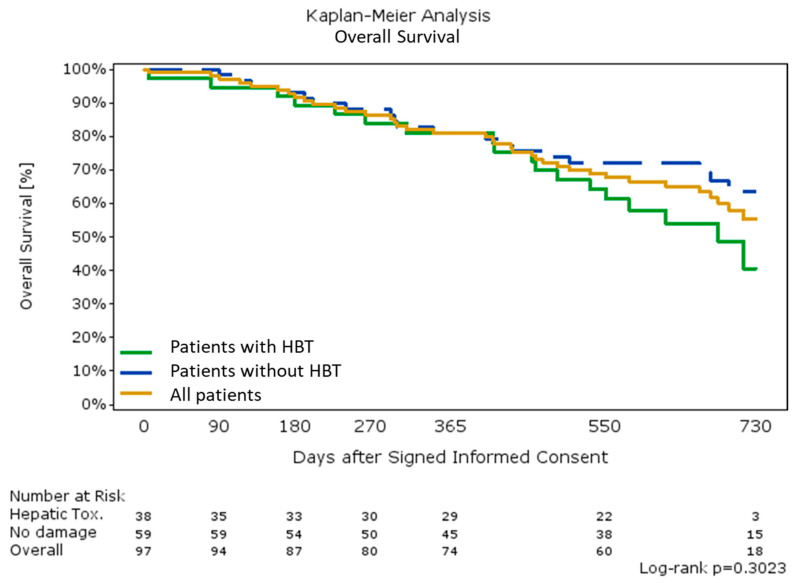
Overall survival (HBT = hepatobiliary toxicities).

**Figure 5 cancers-12-03405-f005:**
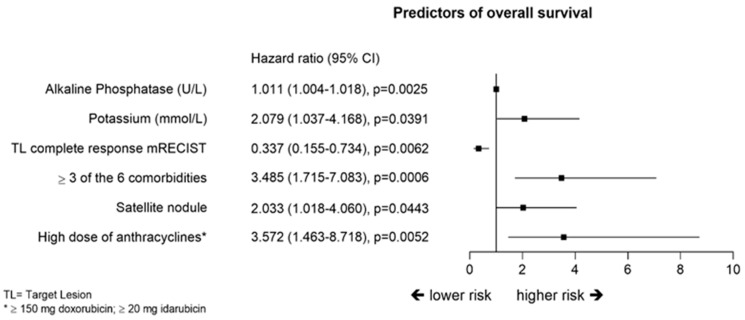
Multivariate analysis for the prediction of overall survival (Cox regression model).

**Table 1 cancers-12-03405-t001:** Baseline characteristics of patients and tumors according to different subsets. Barcelona Clinic Liver Cancer (BCLC); aspartate aminotransferase (AST), alanine aminotransferase (ALT).

		Type of Drug			BCLC Stage		
Variable	Total Population (N = 97)	Doxorubicin (N = 75)	Idarubicin (N = 22)	*p*	BCLC 0&A (N = 46)	BCLC B&C (N = 51)	*p*
Age, years (mean ± SD)	65.9 ± 10.6	66.3 ± 10.1	64.7 ± 12.3	0.62	65.2 ± 11.3	66.6 ± 9.9	0.42
Gender, Male (%)	92.8%	93.3%	90.9%	0.70	87.0%	98.0%	0.04
Cirrhosis (%)	84.0%	91.7%	59.1%	<0.001	88.9%	79.6%	0.40
Child Pugh Score *							
A	90.9%	91.8%	83.3%	0.69	91.67%	94.3%	0.79
B	7.3%	6.1%	16.7%		5.56%	5.7%	
C	1.8%	2.0%	0.0%		2.78%	0.0%	
Ascites	16.5%	4.6%	20.0%	0.08	23.9%	9.8%	0.06
BCLC stage							
BCLC 0	7.2%	9.3%	0.0%	0.14	15.2%	0.0%	0.004
BCLC A	40.2%	42.7%	31.8%	0.36	84.8%	0.0%	<0.001
BCLC B	46.4%	41.3%	63.6%	0.07	0.0%	88.2%	<0.001
BCLC C	6.2%	6.7%	4.6%	0.72	0.0%	11.8%	0.017
Laboratory values							
Alpha-feto protein	279.2 ± 938.9	325.9 ± 1015	22.4 ± 20.9	0.74	43.0 ± 90.4	481 ± 1,251	0.15
AST (IU/L)	52.1 ± 33.8	53.7 ± 35.6	42.4 ± 18.1	0.41	42.8 ± 18.0	61.5 ± 42.7	0.02
ALT (IU/L)	45.8 ± 39.5	46.9 ± 41.1	39.4 ± 28.1	0.52	37.3 ± 26.6	54.5 ± 48.1	0.06
Alkaline phosphatase (IU/L)	127.9 ± 58.8	130.0 ± 61.3	116.5 ± 43.3	0.57	116.2 ± 52.2	141.3 ± 63.7	0.074
Potassium (mmol/L)	4.20 ± 0.49	4.20 ± 0.49	4.20 ± 0.49	0.88	4.11 ± 0.57	4.29 ± 0.38	0.08
Sodium (mmol/L)	139.1 ± 3.6	139.1 ± 3.6	139.2 ± 4.1	0.93	139.8 ± 3.9	138.4 ± 3.3	0.02
Tumor characteristics							
Number of tumors (mean ± SD)	2.3 ± 1.6	2.2 ± 1.5	2.6 ± 2.1	0.83	1.7 ± 0.9	2.8 ± 2.0	0.93
Median (range)	2 (1, 10)	1 (1, 5)	2 (2, 10)	<0.001	2 (1, 9)	2 (1, 10)	
Sum of tumor diameters, mm (mean ± SD)	63.0 ± 41.5	57.2 ± 35.7	84.4 ± 53.9	0.02	39.1 ± 17.6	85.4 ± 45.0	0.001

USD data available for 94 out of 97 patients, * data available for 71 out of 79 patients with cirrhosis.

**Table 2 cancers-12-03405-t002:** Procedure characteristics.

		Type of Drug		BCLC Stage	
Variable	Total Population (N = 97)	Doxorubicin (N = 75)	Idarubicin (N = 22)	BCLC 0&A (N = 46)	BCLC B&C (N = 51)
N° of procedures per patient, Mean ± SD	1.93 ± 1.14	2.08 ± 1.21	1.43 ± 0.60 *	1.76 ± 0.87	2.08 ± 1.32
Median	2.0 (1.0, 7.0)	2.0 (1.0, 7.0)	1.0 (1.0–3.0)	2.0 (1.0, 4.0)	2.0 (1.0, 7.0)
Treatment selectivity, %					
Lobar/selective	23.7	27.7	14.3	21.7	26.5
Super-selective	52.6	48.6	61.9	56.5	56.5
Not reported	23.7	23.6	21.7	21.7	25.5
Bilobar treatment, %	34.0	34.7	28.6	17.4	49.0 *
Bead size 100 µm, %	25.8	29.2	19.1	32.6	19.6
Bead size 200 µm, %	74.2	70.8	80.9	67.4	80.4
Time to last imaging FU, months	7.2 ± 6.5	8.2 ± 6.6	3.7 ± 4.7 *	7.8 ± 6.6	6.9 ± 6.5

* *p* < 0.05 for the comparison between idarubicin and doxorubicin, or between BCLC 0&A and BCLC B&C.

**Table 3 cancers-12-03405-t003:** Adverse events grade ≥ 3 related to LifePearl^®^.

Adverse Events (AE) Type	Total	
	N° AE	N° Patients
**Gastrointestinal disorders**		
Abdominal pain	6	6
Diarrhea	2	1
**General disorders and administration site conditions**		
Fatigue	3	3
General health alteration	1	1
**Infections and infestations**		
Enterobacter cloacae sepsis	1	1
Urinary infection with Proteus Mirabilis	1	1
**Investigations**		
Blood bilirubin increased	1	1
**Skin and subcutaneous tissue disorders**		
Cutaneous lesions on face	1	1
**Vascular disorders**		
Chronic artery occlusion and stenosis of coronary ostium artery	1	1
Hypertension	3	3
False aneurysm of segment V in the liver	1	1
**Total general**	21	13

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
