# Peer review of "Real Life Prospective Evaluation of New Drug-Eluting Platform for Chemoembolization of Patients with Hepatocellular Carcinoma: PARIS Registry"

_cancers, 2020, doi:10.3390/cancers12113405_

Round 1
Reviewer 1 Report
The authors investigated a real-life safety and efficacy of a new drug-eluting platform for chemoembolization of patients with hepatocellular carcinoma. The paper is well written and organized. However, there are several concerns in the manuscript. Tyrosine kinase inhibitor (TKI) and immune-oncology drugs are applied in the clinical practice in some countries. I assume some of the patients received TKI, and the drug affected overall survival in the paper. Please refer to those medicines in the paper and provide how they defined TACE failure.
Abstract
Median (95% CI) overall survival at 2-years was not reached [22.5; n.e.] months, progression-free survival was 13.7 months [11.3; 15.6], and TTUP was 16.7 months [12.7; n.e.]
Does “n.e.” stand for “not estimable”? Please provide a complete sentence. …was not reached estimable (or did not reach estimable)?
Although I guess two survivals mean doxorubicin and idarubicin, please clarify.
TTUP should be spelled out.
Method section
Please describe the way how they choose the drug and the size of the sphere. Did the patients receive TKI after TACE failure? Since the ALP and K are found to be significant factors for survival, their baseline characteristics should be shown in Table 1. What does TL stand for in Figure 5?
Result
What does TL stand for in Figure 5?
Please discuss the finding of the multivariate analysis results in the discussion. Do high-dose anthracyclines indicate a high volume of tumors? Please include more baseline data in Table 1, such as tumor markers and liver function tests.
Author Response
Reviewers comments:
The authors investigated a real-life safety and efficacy of a new drug-eluting platform for chemoembolization of patients with hepatocellular carcinoma. The paper is well written and organized.
Thank you very much.
However, there are several concerns in the manuscript.
- Tyrosine kinase inhibitor (TKI) and immune-oncology drugs are applied in the clinical practice in some countries. I assume some of the patients received TKI, and the drug affected overall survival in the paper. Please refer to those medicines in the paper and provide how they defined TACE failure.
Response:
Indeed, as per current guidelines we presume that many patients have received TKI as first line systemic therapy after reaching TACE untreatable progression (TUTP). However, after patients were resected, transplanted or reached TUTP they were followed, until two years, only for survival. Therefore, details about medication were not captured. Concerning immuno-therapies or combinations with TKI, in the countries where study was conducted, they were not available outside of the clinical trials.
To further clarify this point we have added in methods section following:
Further treatment options of patients that reached TACE untreatable condition were at the discretion of multidisciplinary tumour board.
Patients who were resected, transplanted or reached TTUP were followed only for survival.
Abstract
- Median (95% CI) overall survival at 2-years was not reached [22.5; n.e.] months, progression-free survival was 13.7 months [11.3; 15.6], and TTUP was 16.7 months [12.7; n.e.]
Does “n.e.” stand for “not estimable”? Please provide a complete sentence. …was not reached estimable (or did not reach estimable)?
Response
We agree with reviewer’s comment that the way how it is written currently could be confusing.
This has now been modified and reads as follows:
Survival rates at one and two years were 81% and 66%, respectively while the median overall survival (OS) was not reached.
This has now been modified throughout the manuscript and it is marked in bold
Although I guess two survivals mean doxorubicin and idarubicin, please clarify.
Response:
Actually, there were no two survivals, but instead it was median and in the brackets was 95% Confidence Interval.
- TTUP should be spelled out.
Response:
It is now spelled out: time to TACE untreatable progression (TTUP)
Method section
- Please describe the way how they choose the drug and the size of the sphere.
Response
This was a real-life registry with all procedures and follow-ups done according to routine clinical practice. The size of spheres and drug used were according to operator’s discretion as described in methods section
- Did the patients receive TKI after TACE failure?
Response:
As TKIs are current standard of care for patients who failed locoregional treatment we presume that most of the patients who progressed received TKIs, but as indicated under response N°1 after progression patients were followed only for survival and further treatment options were not captured.
- Since the ALP and K are found to be significant factors for survival, their baseline characteristics should be shown in Table 1.
Response:
Those values of ALP and K and values relevant for liver function were now added in Table 1
- What does TL stand for in Figure 5?
Response:
TL stands for Target Lesion and it was now added in Figure 5.
Result
- What does TL stand for in Figure 5? See response on Point 8.
- Please discuss the finding of the multivariate analysis results in the discussion.
Additional discussion about multivariable analysis is now added to the manuscript and reads as follows:
Potassium and alkaline phosphatase (ALP) were also significant negative predictors of survival. Potassium abnormalities are associated with poorer outcomes, particularly in patients with cirrhosis ascites and renal impairment. ALP, although present in many tissues also indicates proliferation of the tumors and has been described as negative predictor for survival of patients with HCC after hepatic resection [29]. This finding agrees with higher dose of anthracyclines as another negative predictor because higher dose is needed in case of larger or multiple tumours.
- Do high-dose anthracyclines indicate a high volume of tumors?
Yes, indeed, high dose anthracyclines is generally used in patients with higher tumor volumes. It was incorporated in the discussion about multivariate analysis, see point 9.
Please include more baseline data in Table 1, such as tumor markers and liver function tests.
Tumor markers and liver function tests added in the Table 1.
Reviewer 2 Report
Thierry de Baere, et al reported the prospective evaluation of HCC patients receiving new DEM-TACE treatment. This Paris registry enrolled 97 patients with unresectable HCC were treated with LifePearl™ loaded with doxorubicin or idarubicin and the two-year follow up was performed. This study set out to assess tolerance and toxicity, safety as well as tumor response of LifePearl™ loaded with anthracyclines.
Overall, the current clinical trial presents valuable survival information about HCC patients receiving DEM-TACE using LifePearl™ that a high tumor response rate and good safety was observed in HCC patients. Although Adverse events (AE) were reported in 71/97 (73.2%) patients, only one patient with a history of heart insufficiency, the largest tumor load measuring 20 cm, and 65% liver replacement, did not survive the DEM-TACE procedure. The HBT rate (15.5%) observed in this clinical trial was significantly lower than in studies with the first generations of DEM.
This clinical trial also reported the highest response rate, with an objective response rate of 81% and a disease control rate of 99%, for LifePearl™.
However, PFS in the doxorubicin group (15.7 months) was significantly longer than in the idarubicin group (10.4 months) mainly due to patients in idarubicin group have much larger tumors. Therefore, the treatment benefit between doxorubicin and idarubicin cannot be adequately assessed in this study, which is a bit of shame. Also, the sample size is too small to draw a comparison between doxorubicin and idarubicin. With appropriate study design, the previous studies (Guiu et al, and Roth et al,) have shown that HCC patients receiving idarubicin TACE had longer median PFS than patients receiving doxorubicin TACE.
Given that this study was a single-arm prospective study, the survival benefit of using new LifePearl™ in TACE cannot be explored. Nevertheless, this single-arm clinical trial reports a promising response rate of DEM-TACE using LifePearl™ in patients with unresectable HCC.
Minor points
Throughout the manuscript the writing of median overall survival with 95% CI was incorrect.
For example, in line 31, “Median (95% CI) overall survival at 2-years was not reached [22.5; n.e.] months, progression free survival was 13.7 months [11.3; 15.6]…..
Please refer to the recent report of clinical trial IMpassion130 (ClinicalTrials.gov Identifier: NCT02425891 ) to correct it.
Author Response
Throughout the manuscript the writing of median overall survival with 95% CI was incorrect.
For example, in line 31, “Median (95% CI) overall survival at 2-years was not reached [22.5; n.e.] months, progression free survival was 13.7 months [11.3; 15.6]…..
Please refer to the recent report of clinical trial IMpassion130 (ClinicalTrials.gov Identifier: NCT02425891 ) to correct it.
It has now been corrected, please see answer to comment 2.
Round 2
Reviewer 1 Report
Thank you for the invitation to review the manuscript. The authors fulfilled each of the major compulsory revisions and modified the manuscript as requested. I do not have further comments.